# Comprehensive Genomic and Phenotypic Characterization of *Escherichia coli* O78:H9 Strain HPVN24 Isolated from Diarrheic Poultry in Vietnam

**DOI:** 10.3390/microorganisms13102265

**Published:** 2025-09-26

**Authors:** Minh Duc Hoang, Pham Thi Lanh, Vu Thi Hien, Cheng-Yen Kao, Dong Van Quyen

**Affiliations:** 1Institute of Biology, Vietnam Academy of Science and Technology, 18 Hoang Quoc Viet, Nghia Do, Hanoi 100000, Vietnam; minhhg.work@gmail.com (M.D.H.); lanhpt.bio@gmail.com (P.T.L.); vuthihienibt@gmail.com (V.T.H.); 2Institute of Microbiology and Immunology, College of Life Sciences, National Yang Ming Chiao Tung University, Taipei 11221, Taiwan; kaocy@nycu.edu.tw

**Keywords:** avian pathogenic *Escherichia coli*, antibiotic resistance, colibacillosis, Vietnam poultry, virulence factors, whole genome sequencing

## Abstract

Colibacillosis, caused by avian pathogenic *Escherichia coli* (APEC), represents a major threat to poultry production, leading to significant mortality and economic losses. This study aimed to characterize an APEC strain, HPVN24, isolated from diarrheic chickens at a farm in Hai Phong, Vietnam. The strain was investigated through phenotypic assays, antibiotic susceptibility profiling, and whole-genome sequencing using the Illumina platform. HPVN24 exhibited β-hemolytic activity and resistance to trimethoprim, ampicillin, and ciprofloxacin. Whole-genome analysis identified the strain as serotype O78:H9 and sequence type ST23, with a genome size of 5.05 Mb and a GC content of 50.57%. Genome annotation revealed a wide repertoire of genes involved in metabolism, secretion systems, virulence, and biofilm formation. Virulence-associated genes included those related to adhesion, iron acquisition, hemolysin production, and stress response. Analysis predicted multidrug resistance to 18 antibiotic classes, with particularly strong resistance to fluoroquinolones. Phylogenetic comparison demonstrated that HPVN24 clustered closely with O78:H9 strains isolated from poultry in other regions, suggesting potential transmission across populations. These findings indicate that HPVN24 is a multidrug-resistant and highly virulent APEC strain linked to colibacillosis outbreaks in Vietnam and highlight the need for ongoing surveillance, judicious antibiotic usage, and alternative strategies to ensure poultry health and food safety.

## 1. Introduction

Poultry products, particularly broiler chicken meat, represent one of the most widely consumed sources of animal protein globally. In 2023, global chicken meat production exceeded 125 million metric tons from over 75 million chickens, with an average per capita consumption of 17.04 kg/year in 2022 [1,2]. The industry generated a global market value of approximately USD 160.62 billion in 2024 and is projected to reach USD 268.32 billion by 2033 [3].

In Vietnam, poultry farming is a cornerstone of the agricultural sector, with an estimated 559 million poultry (chickens and ducks) producing over 2.33 million tons of meat in 2023 [4]. Notably, most production originates from small-scale, household-run farms [5], which are more vulnerable to infectious disease outbreaks due to inconsistent biosafety practices.

One of the most economically devastating infectious diseases in poultry is colibacillosis, caused by avian pathogenic *Escherichia coli* (APEC) [6]. APEC can infect chickens of all ages and production types, leading to localized and systemic infections such as airsacculitis, pericarditis, perihepatitis, omphalitis, cellulitis, enteritis, and salpingitis [7,8]. Clinical symptoms include respiratory distress, lethargy, anorexia, and increased mortality, with gross lesions on serosal surfaces [9].

The prevalence and composition of the *E. coli* strains remained diverse, though the most reported pathogenic *E. coli* serotypes in chickens and ducks worldwide and in Vietnam include O1, O18, O78, and O111, with O78 being the predominant strain [10,11,12]. In addition, other clinically pathogenic serotypes are also found in Vietnam, such as O111B4, O86B7, O126B16, O55B5, O119B4, 0127B8, and O16B6 [11].

The pathogenicity of colibacillosis is closely linked to the virulence profile of the infecting APEC strain, which can harbor diverse virulence genes associated with adhesion, iron acquisition, immune evasion, and toxin production [13,14]. Current treatment primarily relies on broad-spectrum antibiotics such as aminoglycosides, fluoroquinolones, β-lactams, tetracyclines, and trimethoprim [15,16]. Prevention involves vaccination [17], typically using live-attenuated or inactivated strains. In Vietnam, the only commercial vaccine available is based on the APEC O78 serotype, i.e., the Navetco Poultry Sorphysis and *E. coli* inactivated alum vaccine (Navetco, Vietnam, https://navetco.com.vn/san-pham/tu-huyet-trung-va-ecoli-gia-cam-59.html (accessed on 2 July 2025)).

However, the escalating emergence of multidrug-resistant (MDR) APEC strains has severely compromised the effectiveness of antibiotic therapies [18]. These strains often carry genes that confer resistance to multiple antibiotic classes, including first-line treatments for colibacillosis. Alarmingly, recent data indicate that 62.1% of poultry farmers in Vietnam administer antibiotics on their farms, with 74.9% initiating treatment at the first signs of illness, frequently without prior diagnostic testing or antibiogram analysis [19]. Combined with limited biosecurity protocols, such practices increase the risk of treatment failure, facilitate pathogen spread, and contribute to the growing antimicrobial resistance (AMR) crisis [20]. Despite these challenges, there remains a significant gap in the genomic characterization of APEC strains circulating in Vietnam, especially those with high virulence and MDR profiles sharing the serotype O78 [21].

The present study aims to comprehensively characterize the physiological features, antibiotic susceptibility profile, and whole-genome sequence of *E. coli* strain HPVN24 isolated from chickens with diarrhea syndrome on a broiler farm in Hai Phong Province, Vietnam.

## 2. Materials and Methods

### 2.1. Bacterial Strain

The *E. coli* strain HPVN24 was provided by the Molecular Microbiology Department, Institute of Biology (IB), Vietnam Academy of Science and Technology (VAST). This strain was isolated during an outbreak of diarrhea in a broiler flock in Hai Phong province, Vietnam. The affected flock (approximately 5 weeks old) showed ~30% morbidity, with clinical signs including watery diarrhea, reduced feed intake, stunted growth, and increased mortality (Appendix A). Fecal and intestinal samples were aseptically collected from symptomatic chickens and transported on ice to the laboratory. Isolation was performed on selective agar media, and 15 colonies with *E. coli* morphology were purified and identified by the Matrix-assisted laser desorption ionization-time of flight (MALDI-TOF) mass spectrometry (MS) [22] with MALDI-TOF Biotyper (Bruker, Bremen, Germany) in 2023.

### 2.2. Hemolytic Activity

The hemolysis assay was employed to assess the virulence of the bacterial strains by evaluating their impact on blood cells using a blood agar plate containing 5% sheep blood [23]. *E. coli* exhibiting γ-hemolysis provided by the Molecular Microbiology Department, IB, VAST, was used as a reference. The blood agar plates were incubated at 37 °C for 18–24 h under anaerobic conditions.

### 2.3. Antibiotic Susceptibility

The antibiotic susceptibility and minimum inhibitory concentration (MIC) of the strain were assessed with E-test antibiotic testing strips (bioMérieux SA, Marcy-l’Étoile, France) for Ciprofloxacin (CI), Tetracycline (TC), Doxycycline (DC), Trimethoprim (TS), and Ampicillin (AM) according to the manufacturer’s instructions. The turbidity of bacterial suspensions (approximately 10^8^ CFU/mL) was spread onto the MHA plates, and the strips were placed and incubated at 37 °C for 24 h. The classification of resistance or susceptibility was based on CLSI M100 (2023) guidelines, with additional verification using EUCAST 2023 when necessary. MDR was defined as resistant to at least one agent in three or more antimicrobial classes, according to Magiorakos et al. [24].

### 2.4. DNA Extraction and Genome Sequencing

Overnight bacterial inoculum was centrifuged at 12,000 rpm for 1 minute at 4 °C. The supernatant was removed, and the pellet was used to extract total DNA using the GeneJET Genomic DNA Purification Kit (Thermo Fisher Scientific, Waltham, MA, USA) following the manufacturer’s instructions. The quality of extracted DNA was assessed by electrophoresis on a 1% agarose gel, stained with Ethilium bromide (Merck, Darmstadt, HE, Germany), and observed under UV light. The concentration and purity of the DNA were determined using a NanoDrop Lite (Thermo Fisher Scientific, Waltham, MA, USA), yielding sufficient quality for downstream sequencing with the Illumina HiSeq 3000 platform (Illumina Inc., San Diego, CA, USA).

### 2.5. Genome Assembly

Raw sequencing reads were initially assessed for quality using Falco v1.2.4 + galaxy0 [25]. Adapter sequences and low-quality or duplicate reads were subsequently removed using fastp v0.24.0 + galaxy4 [26]. High-quality reads were assembled de novo using SPAdes [27], and resulting contigs were scaffolded using Ragtag v2.1.0 + galaxy1 [28], with *E. coli* reference genomes χ7122 (GCA_000307205.1) and IMT2125 (GCF_000308975.1) as guides to correct potential misassemblies. The completeness and structural quality of the draft genome were evaluated using Quality Assessment Tool (QUAST) v5.3.0 + galaxy0 [29] and Benchmarking Universal Single-Copy Orthologs (BUSCO) v5.8.0 + galaxy1 [30], employing the *Enterobacterales_odb10* dataset and Prodigal-based gene prediction. Additional comparative pan-genomic analysis was performed using Roary v3.13.0 + galaxy3 [31] to assess gene presence and absence across multiple genome assemblies. All analyses were conducted through the European Galaxy Server (https://usegalaxy.eu/ (accessed on 24 February 2025)) [32].

### 2.6. Genomic Analysis

#### 2.6.1. Multilocus Sequence Typing and Serotyping

Multilocus sequence typing (MLST) was performed using tools from the Center for Genomic Epidemiology (https://www.genomicepidemiology.org/ (accessed on 11 February 2025)) based on established schemes [33,34,35]. Serotyping of O- and H-antigens was conducted using SeroTypeFinder v2.0 [36] and ChTyper v1.0 [37,38] and confirmed with a local run of ECTyper on Ubuntu v24.02 Long-Term Support.

#### 2.6.2. Genome Annotation and Visualization

Genome annotations were conducted using both Prokka [39] and Bakta [40]. The annotated genome was uploaded to Proksee (https://proksee.ca/ (accessed on 25 February 2025)) [41] to generate a circular genome map incorporating GC content, GC skew, Prokka/Bakta annotations, and resistance genes identified via Comprehensive Antibiotic Resistance Database—Resistance Gene Identifier (CARD-RGI) [42].

#### 2.6.3. Functional Genomic Analysis

Clusters of orthologous group (COG) classification was performed using COGclassifier v2.0.0 [43]. Functional pathway annotation and completeness analysis were conducted using the Kyoto Encyclopedia of Genes and Genomes (KEGG) database via EggNog-mapper [44], followed by visualization with KEGGaNOG [45], including heatmaps and bar plots. Visualizations were created using R v4.5 (RStudio v2024.12.1+563) with custom scripts.

#### 2.6.4. Detection of Virulence and Antimicrobial Resistance Genes

Virulence genes were identified using VirulenceFinder v2.0 [38,46,47] and ABRicate v1.0.1 [48], referencing the Virulence Factors Database (VFDB) [49]. Antibiotic resistance genes were predicted using the CARD-RGI [42] web server with “Perfect and Strict” settings and further verified with ABRicate [48] using the ResFinder [50] database. Only gene hits with ≥90% identity were considered. Manual cross-checks with annotation results were performed for validation.

#### 2.6.5. Comparative Genomics and Phylogenetic Analysis

Average Nucleotide Identity (ANI) analysis was conducted using FastANI and visualized using ANIclustermap v2.0.1 [51]. The analysis included 20 *E. coli* clinically sampled strains from Asia, Southeast Asia, the United States, the United Kingdom, Denmark, and Australia, with human Uropathogenic *E. coli* (UPEC) *E. coli* CFT073 serving as the outgroup (Appendix A). Briefly, a total of 22 strains were used for comparative and phylogenetic studies including: (i) Korean isolates (O1K1H7_Korea, O2K1H5_Korea, O78H4_Korea, O78H51_Korea [52]); (ii) China isolates (O2K1H5_IMT5155_China [53], O9H21_China, O76_O8H9_China, O78H9_China); (iii) Pakistan isolates (O1H7_Pakistan, O36H5_Pakistan [54], O109H51_Pakistan [54], O-H5_Pakistan [54]); (iv) Southeast Asia region (SEA): Indonesia (O16H48_Indonesia), Thailand (O82H8_Thailand [55]), Brunei (O-H32_Brunei), Laos (O8H25_Laos [56], Vietnam (n O78H9_HPVN24_VN); (v) Other regions: United Kingdom (UK) (O78H9_IMT2125_UK [57], United States (US) (O78_US) [58], Australia (O78H4_Australia), Denmark (O25H4_Denmark); (vi) Outgroup: CFT073 (CFT073_US) [59]. All strains except for the outgroup were identified as APEC based on either provided metadata from NCBI datasets/bioprojects/biosamples databases or analysis of APEC characteristic virulence gene hits via VirulenceFinder with genes such as *fimH*, *iss*, *fyuA*, *hlyE*, *ompT*, *iutA,* and *iroN.*

Phylogenetic reconstruction was performed using core-genome alignments generated with progressiveMauve v2.4.0 [60]. The alignment was processed using the convertAlignment pipeline [61] to remove uninformative sites and concatenate sequences. A maximum likelihood tree was constructed using RAxML-ng [62] and visualized via Interactive Tree of Life (iTOL) [63].

#### 2.6.6. Pan-Genome and Cross-Strains Genetic Profiles Analyses

The pan-genome was analyzed using the Bacterial Pan Genome Analysis pipeline (BPGA) software v1.3 [64] and Roary v3.13.0 [31] with plotting scripts to construct pan-genome gene accumulation, core/soft-core/shell/cloud gene distribution, and number of new genes across studied strains plots. The BPGA was performed with the Usearch clustering algorithm with a 50% cut-off, and the Roary was performed with default parameters with an 80% cut-off for the BLAST+ search [38] and included script-facilitated plotting.

The heatmaps made use of ABRicate v1.0.1 [48] summaries from VFDB [49] for virulence factors and CARD [42] for antibiotic resistance gene databases. Annotations of gene hits from both databases were manually acquired directly from their representative web servers and visualized with custom scripts using R v4.5 and RStudio.

## 3. Results

### 3.1. Hemolytic Activity of the E. coli HPVN24 Strain

In this study, the *E. coli* HPVN24 not only elaborated a hemolysin but also demonstrated the strongest hemolytic activity (β-hemolysis) among other hemolysin-producing isolates (mainly composed of α-hemolysis), which is responsible for the transparent zone surrounding bacterial colonies on sheep blood agar (Appendix A).

### 3.2. Antibiotic Resistance Profile

The antibiotic susceptibility of *E. coli* HPVN24 was assessed using E-test strips for five antibiotics (Appendix A). No inhibition zones were observed around TS and AM (0 mm), while minimal inhibition was seen around CI (1 mm) and tetracycline TC (1 ± 0.5 mm), suggesting high resistance. The largest inhibition zone was observed with DC (10 ± 0.5 mm in diameter). The MICs clinical breakpoints were recorded for CI (12 µg/mL), TC (96 µg/mL), and DC (11 µg/mL), confirming multidrug resistance with relatively better susceptibility to DC. Regarding CLSI M100 2023 guidelines, DC was indicated to be susceptible, while TS, AM, CI, and TS were classified as resistant based on the breakpoints.

### 3.3. Genome Assembly and Annotation

The draft assembly has a total length of 5,053,087 base pairs (bp) with 712 contigs >= 0 bp, 495 contigs > 200 bp, and 222 contigs >= 1000 bp and was visualized (Figure 1). The N50/L50 values were recorded at 50829/29 with 10X coverage and the GC content of 50.57% with no identifiable N’ gaps. Compared to raw reads, the assembly successfully mapped 99.52% and presented 98% of genes properly paired. Simultaneously, the K-mer-based compliance reached 95.43%, meaning that most of the sequences from the raw reads are incorporated into the draft assembly. The K-mer-based correct length is 93.99%, suggesting a high proportion of the assembly consensus with K-mers derived from raw reads. With the low duplication ratio of 1.002, there was minimal possibility of duplicated reads affecting the quality of the assembly. BUSCO analysis identified 432/440 complete genes (431 single-copy, 1 duplicate), with only 7 fragmented and 1 missing, indicating near-complete representation of core single-copy orthologs.

Furthermore, the annotation with Prokka and Bakta revealed rich genomic features. Prokka annotated 4728 features with 4646 protein-coding sequences (CDS), 77 transfer ribonucleic acids (tRNAs), and 4 ribosomal ribonucleic acids (rRNAs). Compared to Bakta, there were 5022 features with 4636 CDS, 95 tRNAs, 4 rRNAs, 1 transfer-messenger RNA (tmRNA), 198 non-coding RNAs (ncRNAs), 57 ncRNA regions, 19 short open reading frames (sORFs), 3 Clustered Regularly Interspaced Short Palindromic Repeats (CRISPRs), 3 origins of chromosomal replication (oriCs), and 6 origins of transfer (oriTs). The total number of hypothetical proteins from Bakta is significantly lower than that of Prokka, which is 377 versus 1088 proteins. Six protein regions in Prokka lacked Bakta annotation, and 237 hypothetical proteins were unannotated by both tools.

### 3.4. Serotyping and Multilocus Sequence Typing

Serotype analysis identified strain HPVN24 as *E. coli* APEC serotype O78:H9 based on *E. coli*–related genes, including *wzx*, *wzy,* and *fliC* at alleles 3, 6, and 1, respectively. This strain did not have a complete group 2 capsular system to give rise to K-antigen, as it lacked core *cps/kps* genes to form the characteristic outer membrane capsular polysaccharide.

MLST analysis revealed that this strain belongs to sequence type (ST) 23, with recognized housekeeping genes and its associated locus of adenylate kinase—adk (6), fumarate hydratase—fumC (4), DNA gyrase subunit B—gyrB (12), Isocitrate/Isopropylmalate dehydrogenase—*Icd* (1), malate dehydrogenase—mdh (20), adenylosuccinate dehydrogenase—purA (13), and ATP/GTP binding motif protein—recA (7). The *fimH* gene (allele 35) was also present, indicating type 1 fimbrial adhesin.

### 3.5. Functional Genomics Analysis

#### 3.5.1. General Genomic Profiling Based on COG Classification

The functional annotation of the *E. coli* HPVN24 genome using the COG database revealed the distribution of predicted coding sequences across various functional categories (Figure 2). The obtained result showed that the most abundant gene categories were those involved in carbohydrate transport and metabolism (G) and amino acid transport and metabolism (E), comprising approximately 400 and 370 genes, respectively. These were followed by genes associated with cell wall/membrane/envelope biogenesis (M), energy production and conversion (C), transcription, translation, and ribosomal structure and biogenesis (K, J), and inorganic ion transport and metabolism (P)—each category containing more than 200 gene copies.

In contrast, categories such as RNA processing and modification (A), chromatin structure and dynamics (B), nuclear structure (Y), and cytoskeleton (Z) were either absent or minimally represented, which is consistent with the prokaryotic nature of *E. coli*, lacking membrane-bound organelles and complex intracellular architecture. Importantly, approximately 170 genes were assigned to the DNA replication, recombination, and repair (L) category, suggesting a robust genomic capacity for maintaining genome integrity and responding to DNA damage. These results provide a broad overview of the functional potential of the HPVN24 strain and highlight key metabolic and cellular processes that may contribute to its survival and pathogenicity.

#### 3.5.2. Biological Pathways Profiling with KEGG

The KEGG-based functional annotation provided a comprehensive overview of the metabolic and cellular pathways encoded in the *E. coli* HPVN24 genome (Table 1). Among the most complete pathways (completeness score approaching 1.0) were those involved in amino acid metabolism, carbohydrate metabolism, biofilm formation, cell motility, metal ion transport, virulence, host adaptation, and secretion systems. As the completion of each described pathway depends on the availability of its key enzymatic steps, completed pathways are likely to be fully functional. Therefore, these enriched pathways suggest a broad repertoire of functions essential for bacterial survival, colonization, and pathogenicity in host environments.

A detailed examination of secretion systems revealed differential levels of pathway completeness (Table 1). The type II secretion system (T2SS) exhibited the highest level of completeness, supporting its role in translocating folded proteins from the periplasm to the extracellular space, an important mechanism for biofilm matrix exportation. The type IV secretion system (T4SS) showed approximately 75% completeness and is implicated in conjugation and horizontal gene transfer, indicating the potential for genetic exchange and adaptability. The type I secretion system (T1SS) demonstrated 65% completeness and facilitates the direct transport of proteins from the cytoplasm to the extracellular environment.

In contrast, the type VI secretion system (T6SS) was among the least complete. Although T6SS has been linked to APEC virulence, particularly in host colonization and interbacterial competition, its partial absence in HPVN24 may reflect strain-specific adaptations or evolutionary gene loss.

In addition to these dedicated secretion pathways, core translocation systems such as the Sec-SRP (Secretion-Signal Recognition Particle) and Tat (Twin-Arginine Translocation) pathways were also detected, highlighting the presence of conserved protein export machinery. Together, these findings underscore the functional versatility of HPVN24 in protein transport, environmental adaptation, and host interaction.

### 3.6. Virulence Genes

Genomic analysis of the *E. coli* O78:H9 strain HPVN24 revealed a broad spectrum of virulence-associated genes contributing to host colonization, stress adaptation, and immune evasion. The strain encodes a complete Type I fimbrial operon (*fimA–I*, including *fimH* allele 35), facilitating mannose-sensitive adhesion to epithelial cells. Additional adhesins, including *rluD*, *yjhB*, *ecpR*, *fdeC*, and the *yeh* operon, further support host surface attachment.

Genes involved in stress response were also identified, such as global regulators *crp* and *rpoS*, and multiple two-component systems (*CpxA/CpxR*, *BarA/UvrY*, *RstA/RstB*, *and PhoB/PhoR*). Antioxidant defense genes (*oxyR*, *soxR*, *and soxS*) and acid resistance genes (the *gad* family) contribute to environmental resilience. The genome encodes multiple iron acquisition systems, including enterobactin (*entA–F*, *fep*), salmochelin (*iroBDEN*), aerobactin (*iucABCD*), and yersiniabactin (*ybt*), enhancing survival under iron-limited conditions. The presence of *hlyE* suggests hemolytic activity that aids in iron uptake and host cell damage. A wide array of biofilm-related genes was detected, including curli fimbriae (*csgA–G*), cellulose biosynthesis (*bcsA*-*C*, *bcsE*-*G*, *bcsQ*, *bcsZ*), PGA synthesis, *BssS*, and colanic acid-related genes. The *flu* gene encoding Antigen 43 (Ag43) promotes autoaggregation and biofilm formation, regulated by *oxyR*. An extensive group 4 capsule locus (*gfcB–E*, *etp*) was also present, with *gfcC* playing a major role in virulence.

For immune evasion, the *iss* gene enhances serum survival. Surface protein encoded by *nlpI* recruits C4b-binding protein (C4bp), suppressing lectin complement pathways, while *ompW* binds factor H to inhibit the alternative pathway.

Overall, the identified virulence genes underscore HPVN24′s pathogenic potential through multifaceted mechanisms involving adhesion, stress tolerance, nutrient acquisition, and immune system evasion.

### 3.7. Antibiotic Resistance Genes

Comprehensive genome analysis of the *E. coli* O78:H9 strain HPVN24 revealed a wide array of antibiotic resistance genes (ARGs) conferring mechanisms both active via efflux pumps, drug alterations/inactivation and passive resistance with membrane permeability reduction with porins and biofilm. The strain developed insensitivity to 18 classes of antibiotics, for example, fluoroquinolones, diaminopyrimidines, glycopeptides, nucleosides, phenicols, tetracyclines, penicillins, beta-lactams, glycylcyclines, cephalosporins, and macrolides (Table 2). Most antibiotic classes were represented by a lower number of antibiotic alteration/inactivation genes, though the antibiotic efflux pumps remained the key resistance mechanism for producing synergetic effects that develop resistance to multidrugs. The strain had four major pump classes: ATP-binding cassette (ABC, 5 hits), resistance-nodulation-cell division (RND, 24 hits), major facilitator superfamily (MFS, 17 hits), and small multi-drug resistance (SMR) (KpnE/KpnF) antibiotic efflux pumps (Appendix A).

There was an agreement between the resistance genotype and phenotypes as reflected in the potent in vitro resistance against trimethoprim, ampicillin, ciprofloxacin, and tetracycline. The presence of several genes and mutations actively prevented inhibitions aside from exclusion pumps. The trimethoprim-resistant dihydrofolate reductase (DHFR) *dfrA1* prevented binding to bacterial DHFR, while beta-lactamase with the *AmpC* variant *bla_EC-13_* and bacterial porins protected against ampicillin. Remarkably, two mutations in the *gyrA* gene, including p.S83L (tcg→ttg) and p.D87N (gac→aac), were identified, leading to amino acid substitutions of serine to leucine and aspartate to asparagine, respectively. In the *parC* gene, the p.S80L mutation similarly altered serine to leucine. These SNPs are strongly associated with high-level fluoroquinolone resistance, with the *parC* mutation playing a pivotal role in the resistance phenotype [65]. Additionally, the *AcrAB-tolC* with mutations Y137H and G103S in *marA* as well as in *AcrR* were annotated but do not lead to overexpression in *the marRAB* operon and ultimately increase *AcrAB* production [66,67].

On the other hand, *E. coli* strain HPVN24 also conferred resistance through target modifications of undecaprenyl pyrophosphate-related proteins and pmr phosphoethanolamine transferase against peptide antibiotics; the glycopeptide resistance gene cluster and Van ligase targeted glycopeptides; and the KdpD/KdpE efflux system prevented aminoglycoside inhibition. Furthermore, our strain featured the antibiotic resistance gene *APH(3″)-Ib/strA* against streptomycin.

### 3.8. Genomic Comparative and Phylogenetic Analyses

Comparative genomic analysis using Average Nucleotide Identity (ANI) and maximum likelihood phylogenetic inference was performed to assess the relatedness of *E. coli* O78:H9 strain HPVN24 to other reference strains (Figure 3 and Figure 4). The ANI heatmap (Figure 3) showed that HPVN24 shares high genomic similarity (99.5–99.8%) with O78:H9 strains from China (GCF_040932925.1) and the United States (GCF_000332755.1), indicating a close evolutionary relationship among poultry-associated strains of this serotype. Additionally, a 99.5% ANI was observed with the O76/O8:H9 strain from China, despite slight serotype differences.

ANI values decreased to approximately 98.6–98.8% when compared to other avian-pathogenic *E. coli* strains, including O78:H51, O8:H25, O109:H51, O9:H21, O16:H48, and O82:H8, some of which originated from Southeast Asia (Laos, Indonesia, and Thailand), suggesting regional diversification. In contrast, the Korean O78:H4 strain showed lower similarity (96.8–97%) despite sharing the same O-antigen group, highlighting the influence of H-type variation on genomic divergence. The phylogenetic tree (Figure 4) further supported the ANI clustering, placing HPVN24 within a well-supported clade of poultry-derived O78:H9 strains, particularly those from China and the US. Strains such as O1, O2, O25, and the human uropathogenic outgroup CFT073 showed distant relationships, consistent with their distinct host origins and genomic backgrounds. These findings suggest that HPVN24 is part of a globally disseminated, poultry-adapted lineage, with genetic signatures closely aligned to strains circulating in East and Southeast Asia, as well as North America.

### 3.9. Pan-Genome and Genetic Profiles Comparison

Pan-genome analysis revealed a substantial number of cloud genes at 6688, which were observed to be strain-specific genes. Core/soft-core and shell genes were comparable when considering 21 studied strains, implying low evolutionary divergence and conserved compositions across strains (Figure 5). Concerning the gene accumulation plot following Heaps’ law, the acquisition of new genes will continue to expand the gene reservoir but approach saturation with an α-value of approximately 0.22. The similar trend was also observable in the subgroup of strains closed to the HPVN24 strain, i.e., APEC O78/APEC O78:H9 from the US, UK, and China, with an α-value of 0.10. Noteworthy, the openness of all studied strains as well as the selected subgroup implied the possibility to acquire genes and diversification, thus enhancing adaptations to host environments and geographical distribution. This could increase the number of virulence factors and antibiotic resistance genes and ultimately enhance the pathogenicity of *E. coli* isolates.

Considering the strain HPVN24 in the pan-genome (V16/V2), it contributed significantly to the unique gene pool of *E. coli* isolates when placed with representative strains around the world (Figure 6). However, it contributed the least within its subgroup with less than 1000 genes, while the sibling—O78:H9 from China—has over 4000 genes; APEC O78 from the US and UK followed but was comparable with Vietnam’s counterpart.

The virulence factors heatmap indicated the shared genomic profiles with key genes in biofilm formation; iron acquisition systems, including enterobactin, ferric-enterobactin, adhesion, and pilus chaperone; and the outer membrane protein encoding gene *OmpA* (Figure 7A). In addition, the strain O-:H32 from Brunei was isolated from healthy flocks, therefore serving as a reference to distinguish the genes conferring potent pathogenicity found in colibacillosis-causing strains. We discovered the majority of isolates shared other virulent subsets of iron uptake systems with aerobactin, iron receptor *fyuA* and several strains acquired yersiniabactin and salmochelin. Therefore, the diversification of iron scavenging mechanisms can be a determinant factor to consider whether a strain could cause colibacillosis. Overall, the isolate HPVN24 was fundamentally similar to the one from China; however, it lacked *iroC* to completely form the cluster *iroBCDEN* encoded for the salmochelin system and aerobactin profile (*iuc*). The ABRicate program using the VFDB did not confidently detect *iroD and iroN,* as well as the aerobactin iron acquisition system, in the strain from Vietnam, despite their presence in genomic annotations. This implied the isolation could have possessed altered sequences with potential mutated or novel forms, resulting in low coverage. Yet, it is open to deeper investigations.

In consideration of the antibiotic’s resistance heatmap (Figure 7B), the strain HPVN24 from Vietnam continued to showcase greater variations compared to the O78:H9 from China, which has reflected the local antibiotics exposure history. Regarding aminoglycoside resistance, the strain from Vietnam has only *APH(3″)-Ib,* while the similar one in China has several genes belonging to *AAC*, *aad*, *APH*, and *arr* families. Comparable results have been observed with extended-spectrum β-lactamases, which confer resistance to both penicillins and cephalosporins. Intriguingly, it lacked *floR*, *flosA3*, *linG*, *mphA*, *and mrx* genes targeting diaminopyramidines, sulfonamide, tetracycline, macrolide, and several small efflux pumps. Collectively, despite the low copies of genes directly conferring resistance to various drugs, the comprehensive profile of efflux pumps within the strain might still be sufficient to effectively hinder inhibition from antibiotics.

## 4. Discussion

APEC, an extraintestinal pathotype of *E. coli*, causes significant economic losses in poultry production [12,68,69]. Its increasing antimicrobial resistance complicates disease control and poses risks to both animal and human health [12,68]. Epidemiological studies are therefore essential to identify circulating strains and guide effective interventions. In this study, we comprehensively characterized biological and genomic features of the strain HPVN24, isolated from diarrheic chickens in Hai Phong, Vietnam.

Phenotypically, HPVN24 exhibited strong β-hemolytic activity and multidrug resistance (MDR), with notable resistance to ampicillin and tetracycline. However, the strain remained comparatively more susceptible to doxycycline, a commonly used antibiotic in poultry [70], suggesting that doxycycline may still offer therapeutic potential where resistance to other drugs is high. When evaluated against established resistance breakpoints—trimethoprim (≥4 µg/mL), ampicillin (≥32 µg/mL), ciprofloxacin (≥4 µg/mL), and tetracycline (≥16 µg/mL) [71], HPVN24 consistently exhibited elevated resistance levels compared to other APEC strains reported, such as O78 serotype [72] and the APEC O2 strain [73].

Genomically, the HPVN24 genome had 5,053,087 base pairs with a GC content of 50.57% and was identified as *E. coli* serotype O78:H9 and sequence type ST23. Serotype O78, together with O1 and O2, accounts for over 80% of APEC infections [10,12] and is also associated with diverse clinical syndromes across a broad host range, including humans [74]. Functional annotation (COG, KEGG) revealed broad metabolic adaptability, with enrichment in genes involved in transport, carbohydrate utilization, and protein synthesis, supporting HPVN24′s ability to grow and persist under diverse conditions [75,76]. KEGG further identified virulence factors, secretion systems, and a complete biofilm pathway, indicating this strain’s strong potential for host colonization and resistance to environmental stress, consistent with previous reports linking biofilm formation to enhanced pathogenicity and resilience [77].

APEC strains often harbor multiple genes encoding essential functions and, in some cases, carry several allelic variants of these key genes [10]. In the present study, the HPVN24 strain was demonstrated to have multiple well-characterized virulence factors, such as *iss*, *tsh*, episomal/chromosomal *ompT*, *cvaC*, *iucD* [78], and other protectins (*hlyE*, *wzy*), iron acquisition system (*irp1*, *irp2*, *fyuA*), and two-component systems (*BarA/UvrY*, *RstA/RstB*) [75].

Notably, one of the key adhesins identified was FimH35, part of the type I fimbrial operon, known to facilitate strong adhesion to host epithelial cells [79]. Interestingly, this allele is highly similar to *fimH* variants found in the human uropathogenic *E. coli* O25b:H4-ST131 strain, which is associated with increased fluoroquinolone resistance, biofilm formation, and persistent colonization [80]. The *fimH35* may have been acquired via horizontal gene transfer or recombination with UPEC-related strains, thereby strengthening the pathogenic profile of HPVN24. The similarity with UPEC O25b:H4-ST131 raises concern for possible zoonotic transmission [81], emphasizing the need for surveillance at the animal–human interface for One Health implications.

Moreover, the HPVN24 genome encodes key stress response systems, including the CRP–cAMP pathway and sigma factor RpoS, both conserved in *E. coli* O78 lineages [10]. These regulators coordinate global gene expression under stress. Unlike previous reports of reduced RpoS activity in MDR strains [82], HPVN24 appears to retain effective regulation, supported by two-component systems (TCS) that enhance environmental sensing and adaptation [76].

In addition, classical invasion-associated genes such as *hlyA* and other well-characterized APEC invasions [75] were absent from the HPVN24 genome. Nevertheless, its pathogenic potential remains significant due to the presence of *hlyE*, which encodes a pore-forming cytotoxin, also known as cytolysin A (ClyA) and silent hemolysin, locus A (SheA) of *E. coli*, *Salmonella typhi*, and *Shigella flexneri* [83]. *HlyE* lyses erythrocytes and mammalian cells, forming transmembrane pores with a minimum internal diameter of approximately ~25 Å [83], thereby contributing to host tissue necrosis and the deep tissue colonization. During systemic infection, *HlyE* may also facilitate translocation of the bacterium into the bloodstream, compensating for the absence of canonical invasions. Combined with strong adherence and biofilm-forming capacity, HPVN24 is likely capable of triggering persistent, systemic inflammation [75,76]. Notably, *hlyE* is typically induced under environmental stress conditions, including oxygen and glucose limitation [84], and this activity was confirmed by hemolysis under anaerobic cultivation in our study. Likewise, Murase et al., (2012) also demonstrated hemolysin E-derived hemolytic activity of the O55:H7 strain and other *E. coli* lineages (phylogroups A, B1, and B2) after anaerobic cultivation on a washed blood agar plate [85]. Interestingly, beyond its role in pathogenesis, this virulence factor has recently attracted attention for its potential applications in nanopore technology, vaccine development, and tumor therapy [86].

Like other pathogenic *E. coli*, HPVN24 relies on biofilm formation as a key adaptive strategy for host colonization and protection against environmental stress [87]. The strain carried genes for the Group 4 capsular polysaccharide system, which encodes the O-antigen capsule and has been implicated in biofilm excretion [88]. This capsule system has also been described in other virulent APEC strains such as IMT2125 and χ7122 [57]. The combination of biofilm biosynthesis, strong adhesion, and stress resilience suggests that HPVN24 is not only highly virulent but also well protected against both host defenses and antimicrobial agents.

Furthermore, whole genome sequence analysis demonstrated that the HPVN24 genome contained a broad and diverse set of antibiotic resistance determinants, consistent with phenotypic antibiotic susceptibility testing. The resistance pattern observed aligns with previously reported trends in O78 APEC strains [89], though HPVN24 displays an expanded repertoire of resistance mechanisms. A major contributor is the presence of multiple efflux pump systems, notably RND-type transporters such as AcrAB-TolC, which confer resistance to a wide range of antibiotics [90]. Whereas, the RND pumps superfamily has been described to play a critical role in broad drug resistance, notably against beta-lactams [91]. Therefore, the diversified multidrug efflux pump system is thought to be the most important resistance mechanism of HPVN24, which is also supported by a previous study as seen in MDR *E. coli* strains [92]. Nevertheless, the diversification of efflux pump arrays effectively prevents administered drug effects of HPVN24 by reducing intracellular drug accumulation [93,94]. However, as the strain remained moderately sensitive towards doxycycline—a commonly used antibiotic in poultry to effectively treat colibacillosis [95]—the antibiotic could continue to be used for treating colibacillosis caused by HPVN24.

In addition, the finding of doxycycline resistance determinant led to the observation that the strain’s non-specific efflux pumps were only able to be partially excluded. This further supports the notion that the synergic effects of both efflux pumps and the presence of antibiotic modification genes greatly render a potent resistance phenotype; thus, their lack of presence leads to the development of sensitivity [96]. Consequently, with no genes annotated to confer resistance to the tetracycline group, it was expected that the tetracyclines would affect the strain’s growth and survival [97,98]. Our further analysis found additional resistance determinants included *dfrA1* and *bla_EC-13_* against trimethoprim and ampicillin, respectively. The *dfra1* has been regarded to have high prevalence in pathogenic strains [99]. It encoded a modified dihydrofolate reductase (DHFR) enzyme with an altered binding site, thereby leading to the loss of cooperation between trimethoprim and the NADPH cofactor [100]. This prevented the trimethoprim from effectively binding to the enzyme and blocking dihydrofolate to tetrahydrofolate conversion. On the other hand, *bla_EC-13_*, often considered as *AmpC* β-lactamases, reduced susceptibility towards cephalosporins through increasing catalytic activity against the group [101]. Thus, both genes offered extensive resistance for the strain HPVN24. In addition, aminoglycoside 3′-O-Phosphotransferase encoded by *APH(3″)-Ib* catalyzed the phosphorylation of streptomycin for chemical-modified inactivation [102].

Phylogenetically, the results demonstrated that HPVN24 clusters closely with poultry-associated *E. coli* strains from other regions, including China, the US, and the UK. This indicates potential shared virulence factors and especially resistance determinants across geographically distant strains and highlights the global dissemination of highly virulent O78:H9 lineages. The global spread of these resistance determinants can be attributed to variations in antibiotic administration practices across countries [103]. In Vietnam, small-scale farms often use antibiotics such as ampicillin, tetracycline, trimethoprim, ciprofloxacin, doxycycline, amoxicillin, and colistin either prophylactically or at the first signs of disease, typically without diagnostic guidance, thereby promoting the development of resistant strains [104]. Similarly, in northwestern China, amoxicillin was the most commonly reported drug, followed by norfloxacin, ofloxacin, ceftriaxone, and oxytetracycline [103], while nationwide surveys identified tetracyclines and quinolones as the predominant residues in manure [105]. By contrast, the UK and the US enforce stricter regulations, with veterinary oversight of antibiotic use (e.g., penicillin, lincomycin, and sulfonamides) and, in some cases, bans on antibiotic growth promoters [106,107]. Given the ongoing burden of colibacillosis in Vietnam and the limited efficacy of current vaccines and control strategies, these findings underscore the urgent need for locally relevant APEC vaccines. Targeted immunoprophylaxis could significantly reduce antibiotic dependence and improve poultry health and productivity.

According to pan-genome analysis, the HPVN24 strain did not substantially contribute new genes into the gene pool across the studied strains, particularly in comparison with resemblance isolates from the US, the UK, and China. Similarly, the heatmap for virulence factors supports the similarity between HPVN24 and APEC O78:H9 from China but is missing the cluster for aerobactin with only *iroB*, *iroE,* and low coverage < 60% for *iroD*, *iroN* and no *iroC*. As a result, the strain can increase susceptibility to host lipocalin-2 protein, which in turn sequesters the iron siderophores on the bacteria [108]. Ultimately, the incomplete cluster of salmochelin can reduce the virulence [109]. In contrast, the antibiotic genomic profile heatmap suggested that HPVN24 did not have an extensive exposure history to multiple antibiotics, as it was modest compared to the APEC O78:H9 from China.

Overall, the present study reported the genomic and phenotypic characterization of *E. coli* O78:H9 isolated from diarrheic poultry in Vietnam. Given the critical role of the poultry sector in Vietnam’s economy and food system, the identification and genomic characterization of APEC O78:H9 provide valuable data to support more targeted interventions. These include rational antibiotic use, updated vaccine formulations, improved surveillance strategies, and biosecurity policies tailored to the local production context. In addition, although O78 is a well-recognized APEC serogroup, the H9 subtype is poorly understood, particularly within the Vietnamese poultry industry. Notably, HPVN24 does not encode Shiga toxins, but its zoonotic potential and genomic similarity to highly virulent strains from neighboring countries raise significant public health concerns. This study addresses a critical knowledge gap and underscores the urgent need for molecular epidemiological monitoring of emerging APEC strains to safeguard both animal and public health.

Despite these findings, this study has several limitations that should be acknowledged. The phenotype for antimicrobial resistance of the strain HPVN24 only covered five representative antibiotics commonly used in poultry (ampicillin, tetracycline, trimethoprim, ciprofloxacin, and doxycycline) due to limited resources. While comprehensive testing of all predicted resistance genes was not feasible, the genomic analysis provided an extensive overview of the AMR potential. Another limitation is the incomplete resolution of plasmids. While four plasmids (pCol156, pIncFIB, pSE11, and pIncI1-Iα) were detected, their sequences could not be fully assembled using short-read Illumina data. As plasmids serve as key vehicles for resistance and virulence gene transfer, long-read sequencing will be required in future studies to close plasmid sequences and better evaluate their epidemiological and zoonotic relevance.

## 5. Conclusions

This study provides the first comprehensive genomic and phenotypic characterization of *E. coli* O78:H9 isolated from diarrheic chickens in Vietnam. This strain demonstrated strong β-hemolytic activity, environmental resilience, and resistance to multiple clinically relevant antibiotics, along with harboring key virulence determinants associated with adhesion, toxin production, and biofilm formation. Our findings highlight the potential zoonotic risk of O78:H9 and its contribution to recurrent colibacillosis outbreaks in poultry production systems and emphasize the urgent need for the development of effective, locally adapted vaccines for the circulating O78:H9 lineage and the implementation of genomic surveillance programs to monitor the evolution of this clone.

## Figures and Tables

**Figure 1 microorganisms-13-02265-f001:**
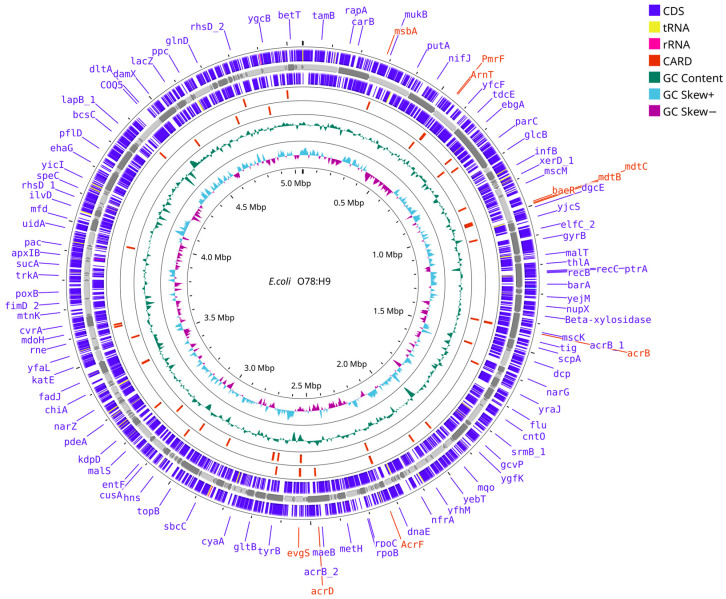
Circular genome map of the *E. coli* HPVN24 strain. Starting from the outermost ring, we show Bakta annotations (+) and (−), Prokka annotations (+), contig backbone, Prokka (−), CARD RGI (+) and (−), GC content, and GC skew in the innermost ring. Moreover, GC content is annotated in dark green; CARD RGI in red; and rRNA, CDS, and tRNA in pink, blue, and yellow. Text colored in red indicated CARD RGI genes while purple was for coding genes.

**Figure 2 microorganisms-13-02265-f002:**
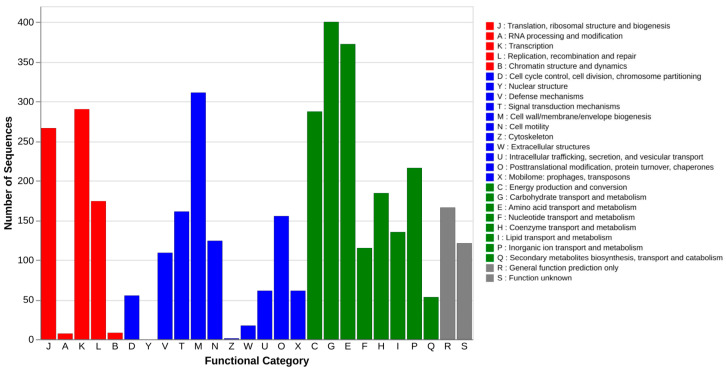
Initial genomic feature classification with the COG database.

**Figure 3 microorganisms-13-02265-f003:**
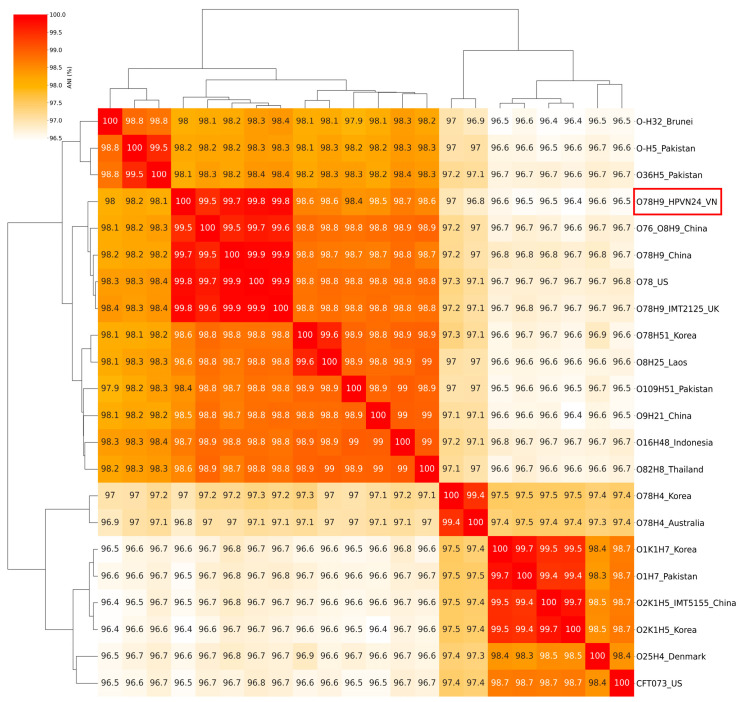
Average Nucleotide Identity heatmap between the *E. coli* HPVN24 and other selected strains. Highly similar strains have the cut-off value > 99% and <= 97% for distant strains. The red box indicated the study’s strain.

**Figure 4 microorganisms-13-02265-f004:**
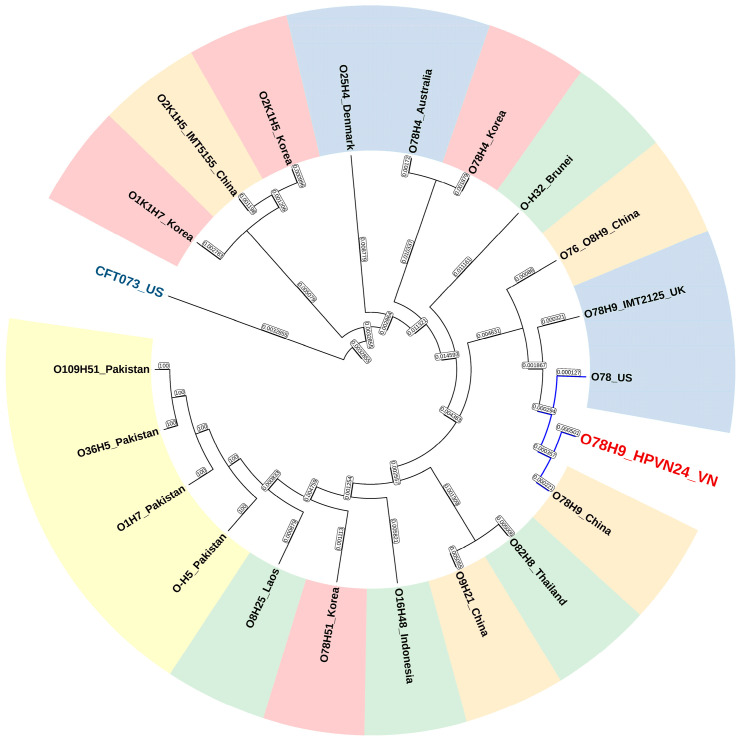
Maximum likelihood phylogenetic tree based on geographical significance and strain relationships. Strains from Korea: red; China: orange; Pakistan: yellow; SEA: green; other regions: blue boxes; outgroup CFT073: dark blue text. The light-blue-colored branches highlight the clade containing the *E. coli* O78:H9 strain HPVN24 and closely related strains.

**Figure 5 microorganisms-13-02265-f005:**
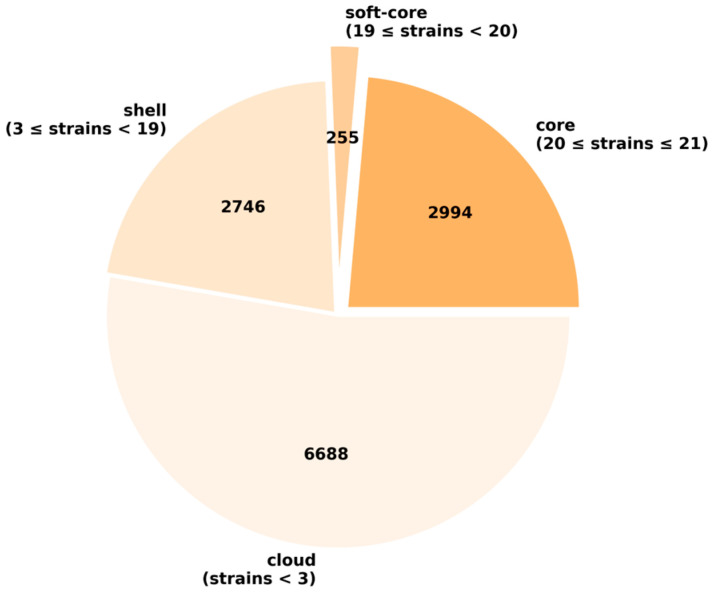
Gene distributions of pan-genome using 21 strains.

**Figure 6 microorganisms-13-02265-f006:**
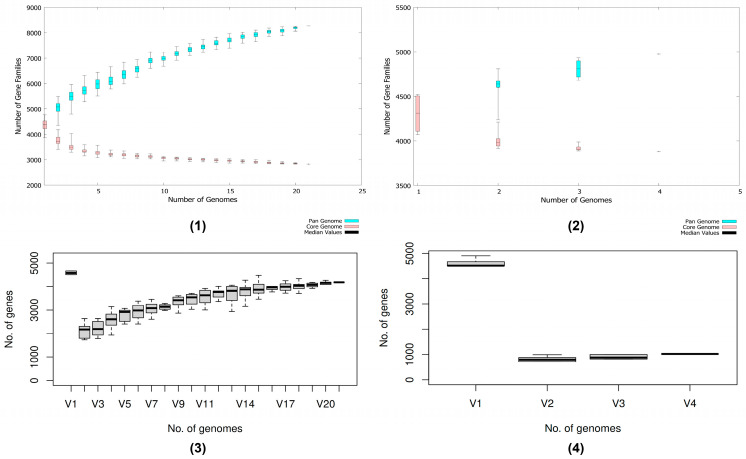
Pan-genome gene accumulation and unique/new gene box plots. (**1**,**3**) plots of 21 strains investigated; (**2**,**4**) plots of strains closely related to the E. coli APEC O78:H9 strain HPVN24. Upper panels (**1**,**2**) show gene accumulation curves with respect to the number of genomes analyzed, where blue and pink sections represent pan-genome and core genome, respectively; black lines represent median values. Lower panels (**3**,**4**) show box plots of novel gene gains, where boxes represent the interquartile range with the median shown as a thick black line within each box. Error bars extend from the boxes to indicate data variability.

**Figure 7 microorganisms-13-02265-f007:**
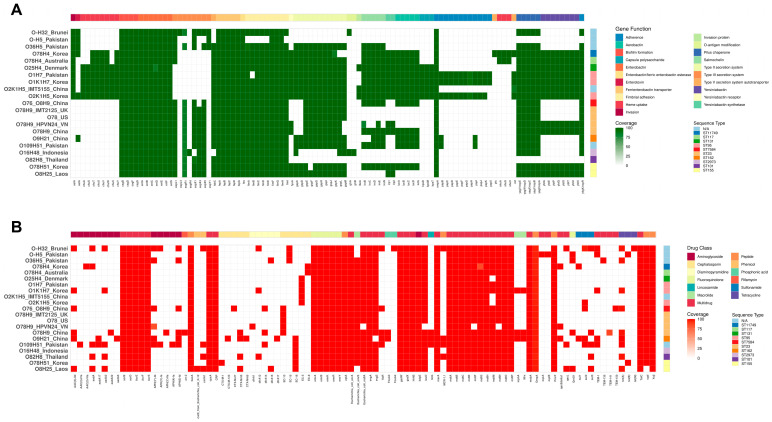
Comparison of multiple strains’ genetic profiles. (**A**) Cross-strain heatmap for virulence factors, gene coverage, and identity cut-off at 80%; the top bar highlights gene functions, and the right bar is for MLST grouping with sequence types. (**B**) Cross-strain heatmap for antibiotic-resistant strains, gene coverage, and identity cut-off at 80%; the top bar highlights drug classes, and the right bar is for MLST grouping with sequence types.

**Table 1 microorganisms-13-02265-t001:** Functional profile and the enriched pathways with KEGG of the *E. coli* O78:H9 strain HPVN24.

Categories	Pathways	Completeness Score	Pathways	Completeness Score
Amino acid metabolism	Alanine	1.0	Lysine	1.0
	Arginine	1.0	Methionine	1.0
	Asparagine	1.0	Phenylalanine	1.0
	Aspartate	1.0	Proline	1.0
	Cysteine	1.0	Serine	1.0
	Glutamate	1.0	Formaldehyde assimilation	0.4
	Glycine	1.0	Threonine	1.0
	Histidine	1.0	Tryptophan	1.0
	Isoleucine	1.0	Tyrosine	1.0
	Leucine	1.0	Valine	1.0
Arsenic reduction	Arsenic reduction	0.75		
Bacterial secretion systems	Sec-SRP	1.0	Type III secretion	0.47
	Twin Arginine targetting	1.0	Type IV secretion	0.75
	Type I secretion	0.66	Type VI secretion	0.11
	Type II secretion	0.92		
Biofilm formation	PGS Synthesis protein	1.0	Colanic acid and Biofilm transcription regulator	1.0
	Biofilm regulator BSS	1.0	Curli fimbriae biosynthesis	0.99
	Colanic acid and Biofilm protein A	1.0		
Carbohydrate metabolism	Alpha-amylase	1.0	Mixed acid: Formate	1.0
	Entner–Doudoroff	1.0	Mixed acid: Formate to CO_2_ & H_2_	0.5
	Glycolysis	0.89	Mixed acid: PEP to Succinate via OAA, malate and fumarate	0.88
	Glyoxylate shunt	1.0	Polyhydroxybutyrate synthesis	0.17
	Mixed acid: Acetate	1.0	Starch/glycogen degradation	1.0
	Mixed acid: Ethanol, Acetate to Acetylaldehyde	1.0	Starch/glycogen synthesis	0.99
	Mixed acid: Ethanol, Acetyl-CoA to Acetylaldehyde (reversible)	1.0	TCA Cycle	0.88
Carbon degradation	Beta-glucosidase	1.0	Diacetylchitobiose deacetylase	1.0
	Bifunctional chitinase/ lysozyme	1.0	Naphthalene degradation to salicylate	0.17
	Chitinase	1.0	Pullulanase	1.0
	D-galacturonate isomerase	1.0	beta-N-acetylhexosaminidase	1.0
Carbon fixation	3-Hydroxypropionate Bicycle	0.24	Gluconeogenesis	0.89
	4-Hydroxybutyrate/ 3-hydroxypropionate	0.2		
Cell mobility	Adhesion	1.0	Flagellum	1.0
	Chemotaxis	0.75		
Genetic competence	Competence-related core components	0.14		
Hydrogen redox	Hydrogen:quinone oxidoreductase	1.0	NiFe hydrogenase Hyd-l	0.99
Metal transporters	Cobalt transporter CorA	1.0	Ferrous iron transporter FeoB	1.0
	Copper transporter CopA	1.0	Nickel ABC-type substrate-binding NikA	1.0
	Fe-Mn transporter MntH	1.0		
Miscellaneous	Anaplerotic genes	0.75	Staphyloaxanthin biosynthesis	0.17
Nitrogen metabolism	DRNA	1.0	Nitrite oxidation	1.0
	Dissimilatory nitrate reduction	1.0		
Oxidative phosphorylation	Cytochrome bd complex	1.0	F-type ATPase	1.0
	Cytochrome o ubiquinol oxidase	1.0	NADH-quinone oxidoreductase	0.84
Sulfur metabolism	DMSO reductase	0.99	Thiosulfate/polysulfide reductase	0.66
	Sulfur assimilation	1.0		
Transporters	Bidirectional polyphosphate	1.0	Transporter: phosphate	1.0
	C-P lyase cleavage PhnJ	1.0	Transporter: phosphonate	0.99
	CP-lyase complex	1.0	Transporter: thiamin	0.99
	CP-lyase operon	0.99	transporter: vitamin B12	0.99
Vitamin biosynthesis	Cobalamin biosynthesis	0.62	Retinal biosynthesis	0.25
	MEP-DOXP pathway	0.99	Riboflavin biosynthesis	1.0
	Mevalonate pathway	0.2	Thiamin biosynthesis	0.91

**Table 2 microorganisms-13-02265-t002:** Resistance genes and their associated mechanisms identified in strain HPVN24.

Resistance Mechanisms	Antibiotic Resistance Ontology (ARO) Category	Resistance Genes	Involved Antimicrobial Classes
Antibiotic target replacements	Perfect	*dfrA1*	diaminopyrimidine antibiotic
Antibiotic target alteration		*PmrF*, *bacA*	peptide antibiotic
Reduced permeability to antibiotic		*marA*	nitroimidazole, macrolide, fluoroquinolone, aminoglycoside, carbapenem, cephalosporin, glycylcycline, penicillin beta-lactam, tetracycline, peptide, aminocoumarin, rifamycin, phenicol, phosphonic acid, disinfecting agents and antiseptics
Antibiotic efflux		*msbA*, *acrA*, *acrB*, *marA*, *mdtE*, *AcrS*, *AcrE*, *evgA*, *H-NS*, *emrB*, *emrR*, *mdtH*, *mdtG*, *cpxA*, *qacEDelta1*	
Reduced permeability to antibiotic	Strict	*SoxS*	fluoroquinolone, monobactam, carbapenem, cephalosporin, glycylcycline, penicillin beta-lactam, tetracycline, rifamycin, phenicol, disinfecting agents and antiseptics
Antibiotic inactivation		*EC-13*	cephalosporin
Antibiotic efflux		*TolC*, *mdtC*, *mdtB*, *mdtA*, *Yojl*, *AcrF*, *acrD*, *emrY*, *emrK*, *evgS*, *mdtP*, *mdtO*, *mdtM*, *kdpE*, *emrA*, *KpnF*, *KpnE*, *CRP*, *gadX*, *rsmA*, *baeR*, *acrR*, *marR*, *SoxS*, *SoxR*	macrolide, fluoroquinolone, aminoglycoside, nucleoside, carbapenem, cephalosporin, glycylcycline, lincosamide, penicillin beta-lactam, tetracycline, glycopeptide, peptide, aminocoumarin, rifamycin, phenicol, phosphonic acid, diaminopyrimidine, disinfecting agents and antiseptics
Antibiotic target alteration		*ArnT*, *eptA*, *vanG*, *ugd*, *acrR*, *marR*, *soxS*, *soxR*	peptide, glycopeptide, fluoroquinolone, cephalosporin, glycylcycline, penicillin beta-lactam, tetracycline, rifamycin, phenicol, disinfecting agents and antiseptics

## Data Availability

The data presented in this study are openly available on Github at https://github.com/DMinhhg/E.coliO78H9-IB-VAST (accessed on 29 June 2025). Genomic data were deposited in the NCBI database under BioProject PRJNA1280482, BioSample SAMN49479598, and GenBank accession JBPJUP000000000 (version JBPJUP010000000) for genome assembly.

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
