# Peer review of "Comprehensive Genomic and Phenotypic Characterization of *Escherichia coli* O78:H9 Strain HPVN24 Isolated from Diarrheic Poultry in Vietnam"

_microorganisms, 2025, doi:10.3390/microorganisms13102265_

Round 1
Reviewer 1 Report
Comments and Suggestions for Authors
This manuscript describes a genotypic characterization (WGS for detection of virulence and AMR genes) and a phenotypic characterization (virulence and AMR) of an APEC (O78:H9) isolate involved in a diarrhea outbreak in poultry in Vietnam.
The introduction is well-written and provides sufficient background to understand the relevance of the study. However, the Materials and Methods section requires further detail to ensure clarity and reproducibility. The results need refinement in certain parts to improve readability, whereas the discussion and conclusion are appropriate.
Remarks
- In the Keywords, replace words that already appear in the title with others that better contextualize the study for improved indexing.
- L48: Enteritis is not reported, similar to what was observed in this study?
- In the Materials and Methods section, the authors need to describe the outbreak from which this APEC originated. How did it occur? What was observed? They should also indicate the sample collection and isolation methodology used.
- L88–92: Reporting colony morphology on different culture media seems unnecessary. Instead, the focus should be on specifying how the isolation that led to the sequenced isolate was performed.
- L97: Do not introduce abbreviations that have not been previously defined. Double-check the entire manuscript.
- Section 2.3: Specify which breakpoints were used for classification as resistant or susceptible. The exact values are not required, but the reference guideline must be indicated. The criteria used for MDR determination should also be defined and referenced.
- Section 2.6.5: Clarify whether all isolates were APEC or specifically O78:H9 to ensure a better understanding of the selection criteria for the phylogenetic analyses.
- Section 3.1: It is recommended to report only hemolytic activity.
- Figure 1: The inclusion of plates does not add value to the results and depicts a well-known methodology. It is suggested to present only the values in the text.
- Figure 4: The figure is very difficult to visualize and interpret. The authors should reconsider the presentation of these results and include additional details in the text.
- Section 3.6: Use virulence genes instead of virulent genes.
- Table 1: Needs reformatting to comply with the journal’s guidelines.
- Table 1: Should also include antimicrobial classes associated with the genes and their mechanisms of action.
- In the AMR section, indicate whether there was agreement between genotype and phenotype.
- Why did the authors not assess phenotypic AMR according to all resistance genes identified in the isolate? This should either be performed or clearly stated as an important limitation of the study.
- Figure 5A: Very difficult to interpret. It is suggested to retain only Figure 5B and, if necessary, report the main results of 5A in the text.
- Data Availability Statement: Include GenBank accession information.
Reviewer 2 Report
Comments and Suggestions for Authors
Dear authors,
This study is important for the surveillance of antimicrobial-resistant (AMR) pathogens with zoonotic potential. Although the english is good but requires minor proofreading.
Methodology: The use of multiple bioinformatics tools and cross-validation of results adds robustness to the analysis. O78:H9 is a globally prevalent and high-impact APEC serotype to contextualize the importance of studying this strain.
Discussion: It is very long. I would be convenient to divide it into topics or sections.
Regarding the presence of the fimH35 allele in the HPVN24 strain, associated with the high-risk uropathogenic clone (UPEC) E. coli O25b:H4-ST131, a more in-depth discussion would be important. For example, discuss: How was this gene acquired? What implications does it have for pathogenesis? and for zoonotic potential?
Paragraphs 574-580 should be discussed further. Is the use of antibiotics the same in different countries?
Conclusion: It is important to emphasize that these findings suggest the need to develop local vaccines with antigens relevant to the circulating O78:H9 lineage. Furthermore, for public health reasons, it would be important to implement genomic surveillance programs to monitor the evolution of this clone.
Figure 1C: Correct the legend to accurately reflect what is shown. The legend should read something like: "(C) Graphical representation of the inhibition zone diameters shown in panel (B)."
Table 1. "Resistance genes and their associated mechanisms identified in strain HPVN24." and remove the internal heading "ARO classifications." The table could be easier to interpret with columns like: Mechanism | ARO Category (Perfect/Strict) | Resistance Genes.
Comments on the Quality of English LanguageThe manuscript is well-written overall, though minor English polishing is recommended
Reviewer 3 Report
Comments and Suggestions for Authors
The manuscript contains the detailed genomic analysis of a freshly isolated avian pathogenic E. coli strain. The study is warranted because of the growing importance of poultry industry within food production, and the growing problem of multidrug-resistance as well as the genomic plasticity of pathogenic E. coli.
The authors performed an impressive array of experiments with state-of-the-art methods, the manuscript in general is excellently written.
One major problem is that the Bioproject, Biosample and Nucleotide sequence records are currently unavailable at NCBI – maybe their accessibility was not set for ‘public.’ Therefore, the findings cannot be verified. The authors should make these data accessible for review.
Another question regarding the genomic findings is if the authors have identified any plasmids? If yes, its nucleotide sequence should also be made available.
Some minor comments:
Table 1: the abbreviation ARO should be explained here, as it only gets explained in the legends of Supplementary Materials
L301: Virulence genes instead of ‘virulent genes’
L359: ‘our strain gained’ – do the authors suggest that it was the result of horizontal gene transfer?
Figure 5A: it should be available in a larger version, or could be moved to Supplementary material.
Figure 6B: same concern as with Fig. 5A
Figure 7: it would be easier / more elegant to refer to the strains with their proper designations, perhaps with a separate column for the serotypes
L425: see my remark for Fig. 7, it would be easier to refer to the strains with a designation, instead of circumscribing it
L442: I would reformulate this sentence, ‘findings are also found’ sounds a little awkward
L556: ‘the finding of doxycycline…’ the words ‘resistance determinant’ are missing here
L584: see my remark for Fig. 7.
L588: supports instead of ‘support’
L592: virulence instead of ‘pathogenicity’, if we talk about reduced capabilities, but still pathogenic status
L595: was instead of ‘did’
Table S1: The headings are a bit misleading and confusing.
The column ‘strain’ does not contain strain identifiers, only the serotypes. Please correct it, and list the strain designations in a separate column.
The column title ’Ascension ID’ is a bit confusing. These are NCBI RefSeq identifiers. Please correct it.
If there are references for any of the strains, please list them in a separate column and in the reference list as well. CFT073 is a historical reference strain with a published reference, so this applies for this strain at least, and it would be more appropriate to include its reference in the main body text as well (L158).
Supplementary Figure S3: the colouring is helpful, but a table with larger font instead would be more readable
Comments on the Quality of English LanguageThe manuscript in general is written in excellent English, however, some parts of the manuscript would benefit from a language editor program, of which there are plenty freely available ones (Paperpal or any other AI-based tools), to clean the text from minor language errors.
Round 2
Reviewer 3 Report
Comments and Suggestions for Authors
The authors have performed all the indicated changes to the manuscript. The genome assembly of the strain, though is still unavailable in Genbank. The authors have made it available through Github, hopefully the Genbank entry becomes accessible until publication.